# Cholesterol-Lowering Effect of Polysaccharides from *Cyclocarya paliurus* In Vitro and in Hypercholesterolemia Mice

**DOI:** 10.3390/foods13152343

**Published:** 2024-07-25

**Authors:** Yang Zhang, Lei Zeng, Kehui Ouyang, Wenjun Wang

**Affiliations:** 1Jiangxi Province Key Laboratory of Animal Nutrition, College of Animal Science and Technology, Jiangxi Agricultural University, Nanchang 330045, China; jiushixiaoyang77@sina.com; 2Key Lab for Agro-Product Processing and Quality Control of Nanchang City, College of Food Science and Engineering, Jiangxi Agricultural University, Nanchang 330045, China; zenglei117@gmail.com

**Keywords:** *Cyclocarya paliurus* polysaccharides, hypercholesterolemic, antioxidant, intestinal flora

## Abstract

In this study, a new component of *Cyclocarya paliurus* polysaccharides (CPP20) was precipitated by the gradient ethanol method, and the protective effect of CPP20 on hypercholesterolemia mice was investigated. In vitro, CPP20 had the ability to bind bile salts and inhibit cholesterol micelle solubility, and it could effectively clear free radicals (DPPH•, •OH, and ABTS+). In vivo, CPP20 effectively alleviated hypercholesterolemia and liver damage in mice. After CPP20 intervention, the activity of antioxidant enzymes (SOD, CAT, and GSH-Px) and the level of HDL-C in liver and serum were increased, and the activity of aminotransferase (ALT and AST) and the level of MDA, TC, TG, LDL-C, and TBA were decreased. Molecular experiments showed that CPP20 reduced cholesterol by regulating the mRNA expression of antioxidation-related genes (SOD, GSH-Px, and CAT) and genes related to the cholesterol metabolism (CYP7A1, CYP27A1, SREBP-2, HMGCR, and FXR) in liver. In addition, CPP20 alleviated intestinal microbiota disturbances in mice with hypercholesterolemia and increased levels of SCFAs. Therefore, CPP20 alleviates hypercholesterolemia by alleviating oxidative damage, maintaining cholesterol homeostasis, and regulating gut microbiota.

## 1. Introduction

Cholesterol is an important factor in the maintenance of cellular homeostasis, and it is synthesized in the liver (about 700–900 mg/day) and ingested through diet (about 300–500 mg/day) [1]. An excessive consumption of foods rich in saturated and trans fats will increase cholesterol levels and lead to hypercholesterolemia. Hypercholesterolemia is characterized by high levels of total cholesterol (TC), low-density lipoprotein cholesterol (LDL-C), or triglycerides (TG), as well as a decrease in high-density lipoprotein cholesterol (HDL-C) [2]. Hypercholesterolemia often leads to atherosclerosis, coronary artery disease, and a range of cardiovascular and cerebrovascular diseases. Statins are commonly used as cholesterol-lowering drugs, but they may cause adverse reactions, such as liver and kidney damage, rhabdomyolysis, elevated blood sugar-induced diabetes, etc., and they are not considered to be an ideal long-term treatment [3]. Therefore, it is important to find new effective drugs and new treatment strategies for preventing or alleviating hypercholesterolemia.

Systemic cholesterol homeostasis is a strictly regulated process involving de novo biosynthesis, dietary cholesterol absorption, biliary clearance, and excretion, and the liver is the main organ to maintain cholesterol homeostasis [4]. The metabolism of cholesterol in the body mainly through the following ways: 1. Cholesterol is transformed into cholesterol derivatives with important physiological functions, such as Vitamin D, sex hormones, and adrenocortical hormones; 2. Cell-membrane formation; 3. Esterification; 4. Excretion in feces; 5. Cholesterol is hydroxylated and degraded to produce bile acids (BAs) [5]. BAs are steroid substances synthesized by the liver from cholesterol. Various bile salts (BS) formed by BAs are the main components of bile, and the conversion of cholesterol into BAs accounts for 80% of total metabolism. There are two main pathways for BA synthesis in hepatocytes: The classical pathway with cholesterol 7α hydroxylase (CYP7A1), and the alternative pathway with sterol 27 hydroxylase (CYP27A1) [6]. When cholesterol levels are high, cholesterol is broken down into BAs and stored in the gallbladder. They are then released into the duodenum and proximal jejunum after cholecystokinin stimulation. After entering the intestine, a part of the BAs is absorbed by the mucosa and returned to the liver to complete the hepatoenteric circulation. In contrast, the unabsorbed parts are degraded into fecal matter by intestinal microorganisms and expelled [7]. Promoting the conversion of liver cholesterol to BAs is an important method to reduce cholesterol, which is of great significance for the prevention and treatment of hypercholesterolemia.

Intestinal flora is related to digestion, nutrient metabolism, immune function, etc., and is important in body health and physiological metabolism. Diet is one of the most critical factors in the change of gut microbiota structure, and the functional components in food can regulate the growth and metabolic activities of gut microbiota, thus affecting the microbial composition. Under normal circumstances, the gut flora and the host benefit from a mutual dynamic ecological balance. Once this balance is disrupted, pathogenic bacteria can cause infectious diseases, metabolic syndromes (such as hyperlipidemia and high blood pressure), and certain neurological disorders [8,9]. Studies have shown that a high-fat diet disrupts the body’s original microbial balance and induces more conditioned pathogens, such as Betaproteobacteria, *Clostridium_bolteae*, *Desulfovibrio*, and *Enterobacter_cloacae* [10,11,12]. In addition, intestinal flora participates in and regulates host metabolism by producing metabolites and short-chain fatty acids (SCFAs) to alleviate hypercholesterolemia [13]. SCFAs can stimulate the proliferation and differentiation of intestinal epithelial cells and help maintain the mineral balance of intestinal epithelial cells and the absorption of iron, calcium, and magnesium. Common SCFAs include acetic acid, propionic acid, and butyric acid, while other SCFAs such as isobutyric acid, valeric acid, and isovaleric acid are relatively low in the colon [14].

Polysaccharides are essential biomacromolecules in living organisms, playing important roles in life activities such as cell–cell connections, signaling pathways, cell adhesion, and immune system molecular recognition. In recent years, more and more studies have reported the isolation and identification of bioactive polysaccharides from natural sources, indicating that polysaccharides have a variety of biological functions, including anti-hypercholesterolemic activity [15,16,17]. *Cyclocarya paliurus* is a species native only to China, and its leaves are used in medicinal preparations in traditional Chinese medicine, as well as an ingredient in functional foods [18,19]. Previous studies have shown that polysaccharides from *Cyclocarya paliurus* leaves can regulate lipid metabolism, relieve hyperlipidemia, and relieve diabetes, but its effect on hypercholesterolemia has not been reported [18,19,20]. Therefore, this study attempted to explore the effect of *Cyclocarya paliurus* polysaccharides on hypercholesterolemia through in vivo and in vitro experiments to explore the mechanism of action.

## 2. Materials and Methods

### 2.1. Materials and Reagents

*Cyclocarya paliurus* leaves were purchased from Nanchang, Jiangxi Province, China. The diagnostic kits used for investigating TC, TG, HDL-C, LDL-C, SOD, GSH-Px, CAT, ALT, AST, and MDA were purchased from Nanjing Jiancheng Bioengineering Institute (Nanjing, China). Standards of SCFAs (acetic acid, propionic acid, butyric acid, valeric acid, isobutyric acid, and isovaleric acid) were purchased from Shanghai Aladdin Biochemical Technology Co., Ltd. (Shanghai, China). Other reagents and chemicals used in this work were of analytical grade and are supplied by local chemical suppliers in China.

### 2.2. Preparation of Polysaccharides from Cyclocarya paliurus 

The *Cyclocarya paliurus* leaves were dried, crushed, and screened by 40 mesh, then soaked with petroleum ether to remove the fat-soluble impurities. The treated powder was soaked in distilled water at 85 °C and extracted three times under the condition of solid–liquid ratio 1:10 (*w*/*v*) with ultrasonic assistance. The extracts were combined, concentrated, and precipitated with 95% ethanol. The protein was removed from the extract by Sevage method, and the crude polysaccharide (CPP) was obtained by freeze-drying after dialysis. The components obtained after precipitation with 20% and 40% ethanol by volume are named CPP20 (21.77 g) and CPP40 (7.23 g).

### 2.3. Characterization of Polysaccharides

#### 2.3.1. Chemical Composition and Structural Analysis

The chemical composition was determined by reference to previous studies with some modifications [21]. The total carbohydrate content, uronic acid, and protein content were estimated by phenol-sulfuric acid method, m-hydroxy-biphenyl method, and coomassie brilliant blue method, respectively.

#### 2.3.2. Analysis of Monosaccharides Composition

The monosaccharide composition was determined using a gas chromatography manufactured in the United States by Agilent Technologies (Model: 19091J413, Hp-5 capillary column, size: 30 mm × 0.25 mm × 320 µm, Santa Clara, CA, USA). Then, 5 mg of the sample were taken and put into ampoules, then 10 mL of 2 M trifluoroacetic anhydride were added and hydrolyzed at 120 °C for 3 h. The hydrolyzed solution was blown dry and dissolved, and the supernatant was taken for analysis. The identification and quantification of monosaccharides was achieved using the Dionex Carbopac™PA 20 column (size: 3 μm × 150 mm, Shanghai, China). In this process, helium is used as a carrier gas and the flow rate is set to 1 mL/min. The temperature of the injector was maintained at 250 °C, and the column temperature was gradually increased from 120 °C to 250 °C at a rate of 3 °C/min and maintained for 5 min.

#### 2.3.3. Determination of Molecular Weight (MW)

Agilent 1260 HPLC system (Agilent, CA, USA) equipped with SHODEX KS-804 and KS-802 columns (8 mm × 300 mm) and refractive index detector using high-performance gel permeation chromatography and Agilent GPC software (Version: A.02.01) was used for detection and analyzation of molecular weight.

#### 2.3.4. Scanning Electron Microscopy (SEM)

The CPP20 powder was fixed with double-sided adhesive, sprayed with gold, and the morphology of CPP20 was recorded with QUANTA 250 scanning electron microscope (FEI, Boston, MA, USA) under 15 kV and vacuum.

#### 2.3.5. Fourier Transform Infrared Spectra (FT-IR) Analysis

The 1 mg CPP20 powder was mixed with 100 mg KBr powder, and the 1 mm flake was prepared. The wavelength range of 4000–400 cm^−1^ was recorded by infrared spectrometer.

### 2.4. Bile Acid Binding Assay and Cholesterol Micelle Solubility Inhibition Capacity

Standard curve for preparation of sodium taurocholate, sodium glycocholate, and sodium cholate was based on previous reports [22]. The pH environment and cholate binding of human gastrointestinal tract were simulated by standard curve: simulate the environment of human stomach acid, adjust pH to 2.0 ± 0.1, simulate digestion for 2 h at 37 °C. Adjust the pH to 7.4 ± 0.1, inject 4 mL of BS solution, and oscillate for 2 h at 37 °C. Then the mixture was centrifuged, the supernatant was taken for determination, and the BA binding rate was calculated according to Equation (1).
BAs binding assay = (1 − C_1_/C_0_) × 100%(1)

C_1_: Content of BAs in the sample reaction solution, mmol/L

C_0_: Content of BAs in reaction solution without the sample, mmol/L

The cholesterol micellar solution containing cholesterol and oleic acid at 5 mmol/L, lecithin and sodium taurine cholate at 10 mmol/L, and sodium chloride at 132 mmol/L was prepared and equilibrated at 37 °C for 12 h. Add the sample reaction and calculate the inhibition rate of cholesterol micelle solubility according to (2).
Cholesterol micelle solubility inhibition rate (%) = (1 − C_1_/C_0_) × 100%(2)

C_1_: Content of cholesterol in the sample reaction solution, mg/mL

C_0_: Content of cholesterol in reaction solution without the sample, mg/mL

### 2.5. Antioxidant Activity In Vitro

The DPPH•, •OH, ABTS+ scavenging activities and reducing power were measured according to the reported method with several modifications [21]. Polysaccharide solutions with concentrations of 0.05, 0.1, 0.15, 0.2, 0.25, and 0.3 mg/mL were mixed and reacted with freshly prepared DPPH•, •OH, ABTS+, and reducing-power reaction reagents, respectively, and VC was used as positive control.

### 2.6. Animal and Experimental Design

This experiment used C57BL/6J male mice, weighting 18.0~20.0 g, provided by Hunan SJA Lake Jingda Laboratory Animal Co., LTD. (Changsha, China) with the license number SCXK (Xiang) 2022-0004. All experiments were conducted in strict accordance with the guidelines of the Chinese Society of Experimental Animals and the Animal Health and Use Committee of Jiangxi Agricultural University (JXAUA01) and approved by the Institutional Animal Care and Use Committee of Jiangxi Agricultural University (Approval ID: JXAULL-2017003). After adaptive feeding for 7 days, they were randomly divided into two groups: normal group (NC) and model group (8 and 40 mice were included, respectively). The NC received a normal diet, while the model group received a high-fat and high-cholesterol diet. After 5 weeks, the blood levels of TC, TG, LDL-C, and HDL-C were measured. The results showed that the hypercholesterolemia model was successfully established. The mice successfully modeled were randomly divided into five groups with eight mice: model group (HFHC), low-dose polysaccharide group (CPP20L), medium-dose polysaccharide group (CPP20M), high-dose polysaccharide group (CPP20H), and positive drug group (SIM). NC and HFHC were administered distilled water, while the other groups were administered simvastatin 10 mg/kg (SIM), CPP20 100 mg/kg 200 mg/kg, and 400 mg/kg (CPP20L CPP20M, CPP20H) solution, respectively. The NC was fed ordinary diet, and other groups were fed high-fat and high-cholesterol diet. Gavage is performed once a day between 9:00 and 10:30 a.m. In the feeding process, the temperature is kept at 18–22 °C, the humidity is between ~50–60%, and the light and dark alternate light is used for 12 h. The ingredients of high-fat and high-cholesterol feed and ordinary feed are shown in Table 1. During the experiment, the weight of the mice was weighed. After ether anesthesia, the eyeballs were removed and blood was taken, while the kidney, spleen, liver, and cecum contents were taken and stored for testing.

#### 2.6.1. Determination of Organ Indexes

After killing the mice, the liver, spleen, and kidneys were removed and weighed immediately. The organ index is calculated as follows:Organ index (mg/g) = organ weight (mg)/body weight (g)(3)

#### 2.6.2. Determination of Liver Index and H&E Staining of Liver

The liver tissue was fixed with 4% paraformaldehyde, and the pathological changes of liver tissue were observed under optical microscope.

#### 2.6.3. Biochemical Analysis

The liver tissue was prepared into 10% liver homogenate with sterile normal saline at 4 °C. Then the homogenate centrifuged at 3500 rpm for 15 min at 4 °C, and the supernatant was collected to determine ALT, AST, SOD, CAT, MDA, TC, TG, LDL-C, HDL-C, and TBA levels.

#### 2.6.4. RNA Isolation and RT-qPCR

The procedure for detecting mRNA levels of CYP7A1, CYP27A1, SREBP-2, HMGCR, FXR, SOD, CAT, and GSH-Px was generally performed according to the reported method [22]. Total RNA was isolated from liver tissue by TransZol UP Plus RNA kits (ER501, TransGen, Beijing, China) with frozen homogenizer at 4 °C. After total RNA concentration normalization, cDNA was synthesized using One-Step gDNA Removal and cDNA Synthesis SuperMix (AT311-03, TransGen, Beijing, China). Real-time PCR system (Thermo, Waltham, MA, USA) was used to perform qPCR according to the instructions of TranStart Tip Green PCR SuperMix kit (AQ141-02, TransGen, Beijing, China). The result is calculated by method 2^−ΔΔCt^. The primer sequences (Table 2) used as follows were synthesized by Beijing Genomics Institute (Beijing, China) Co., Ltd. The qPCR reaction procedure is as follows: 94 °C for 30 s, followed by 45 cycles of 94 °C for 5 s, then 60 °C for 30 s.

#### 2.6.5. Western Blot Assay

In order to explore the mechanism of cholesterol-lowering effect of CPP20, we detected the protein levels of CYP7A1, SREBP-2, and HMGCR with β-actin as the control [22]. Radio-Immunoprecipitation Assay (RIPA) lysis buffer (4 °C) and liver tissue were processed in a frozen grinder to extract protein. The treated homogenate was bathed in ice for 20 min and centrifuged at 4 °C and 10,000 rpm for 10 min. The obtained supernatant was added to the sample buffer and denatured by heating at 100 °C for 10 min. An equivalent amount of protein (10 μL/lane) separated by 12% SDS-PAGE was transferred to a polyvinylidene difluoride membrane (PVDF, Millipore Co., Lincoln Park, NJ, USA). The membrane was incubated with the corresponding target primary antibody overnight (4 °C), washed with TBST, and incubated with the secondary antibody for 2 h (25 °C). Finally, the washed PVDF was added to the chemiluminescence tetrahydrochloride (ECL) reagent and imitated using the Gene Genius Bioimaging System (SYNGENE Co., Cambridge, UK). The Image J software (Version: 1.48v) was used for the determination of all images.

#### 2.6.6. Analysis of the SCFAs

After the mice were humanely killed, the cecal contents were collected and rapidly placed in liquid nitrogen and finally stored at −80 °C to be tested [23]. The concentration of SCFAs was determined by gas chromatograph (GC-2014 Shimadzu Corporation, Kyoto, Japan). The injection volume was 0.4 μL and the injector temperature was kept at 220 °C. The initial temperature was 110 °C/30 s, and it rose to 120 °C at a constant rate of 10 °C/min, held for 4 min, and then continued to rise to 150 °C within 3 min, where the split ratio and flow rate were controlled at 20:1 and 2.5 mL/min, respectively.

#### 2.6.7. Analysis of the Gut Microbiota

The cecal contents of mice were removed and stored at −80 °C until detection. The E.Z.N.A.^®^ Stool DNA Kit (D4015, Omega, GA, USA) was used for isolated total DNA of gut microbiota. The V3–V4 region of bacterial 16S rDNA was amplified and sequenced using the Illumina MiSeq platform (San Diego, CA, USA). The UPARSE (version 9.2.64) pipeline was applied to cluster clean tags into operational taxonomic units (OTUs) with a similarity ≥97%. The sequence data were stored in the NCBI Sequence Read Archive database and analyzed on the Personalbio Cloud platform, a free online platform.

### 2.7. Statistical Analysis

The values were expressed as mean ± standard deviation (M ± SD). The results were analyzed by one-way ANOVA and Duncan’s test with SPSS 20.0 software. *p*-values of less than 0.05 were considered to be statistically significant.

## 3. Results

### 3.1. Chemical Composition and Structural Analysis of Cyclocarya paliurus Polysaccharides

The chemical compositions of CPP, CPP20, and CPP40 are shown in Table 3. Compared with CPP, the polysaccharide components (CPP20 and CPP40) obtained by fractional alcohol precipitation increased the content of total carbohydrates and uronic acid but decreased the protein content. The content of total carbohydrates and uronic acid of CPP20 is the highest, and the protein content of CPP40 is the lowest.

### 3.2. Antioxidant Activity of Cyclocarya paliurus Polysaccharides

The free radical scavenging ability of *Cyclocarya paliurus* polysaccharides was analyzed, and the results were presented in Figure 1. The CPP20 has the strongest ability to clear •OH and DPPH• in the experimental concentration range and has the highest reducing power.

### 3.3. Cholesterol-Lowering Activity of Cyclocarya paliurus Polysaccharides In Vitro

The BAs’ binding rate and cholesterol micelle solubility inhibition capacity in vitro of *Cyclocarya paliurus* polysaccharides are shown in Figure 2. The BAs’ binding rate and the inhibition rate of cholesterol micellar dissolution of the sample were dose-dependent in the experimental concentration range. Among them, CPP20 has the best in vitro cholesterol-lowering effect, and CPP has the worst effect, so we chose CPP20 as the material for the follow-up experiment in vivo.

### 3.4. The Structure of CPP20

The CPP20 structure is shown in Figure 3. CPP20 is a homogeneous polysaccharide with a molecular weight of 56.282 kDa. The monosaccharide composition of CPP20 is glucose (Glc), arabinose (Ara), galacturonic acid (GalA), galactose (Gal), rhamnose (Rha), *N*-Acetyl-D glucosamine (NAG), glucuronic acid (GlcA), xylose (Xyl), glucosamine hydrochloride (GlcN), fucose (Fuc), and galactosamine hydrochloride (GalN), with molar percentage of 0.296:0.214:0.201:0.170:0.036:0.025:0.020:0.018:0.010:0.008:0.003. The surface morphology of CPP20 (Figure 3) is a porous honeycomb with an irregular and rough surface, which may be affected by ethanol during extraction [24]. The characteristic absorption of CPP20 was determined by FT-IR spectroscopy. Between 3600 cm^−1^ and 3200 cm^−1^, a widely extended strong peak was observed, which is caused by the stretching vibration of -OH and is typical of polysaccharides. In addition, a weaker peak can be observed near 2900 cm^−1^, which is caused by the stretching vibration of C-H. In addition, the absorption peak appears at around 1640 cm^−1^, which is caused by the presence of bound water. The absorption band caused by the C-O-C stretching vibration belonging to the sugar ring can be observed in the range of 1000 to 1200 cm^−1^.

### 3.5. Antioxidant and Cholesterol-Lowering Effects of CPP20 In Vivo

#### 3.5.1. Effects of CPP20 on Body Weight, Organ Indexes, and Liver Histological Changes of Mice

The body weight of mice during the experiment is shown in Table 4. Compared with the NC, the weight gain of the mice with a high-fat and high-cholesterol diet was significantly increased (*p* < 0.05). After 5 weeks of intervention, simvastatin and CPP20 significantly reduced body weight in hypercholesterolemic mice (*p* < 0.05), and the effect of CPP20M was similar to that of simvastatin, both reducing body weight to the NC level.

The organ indexes of mice are shown in Table 5. Liver and spleen indexes of HFHC were significantly higher than those of NC (*p* < 0.05). After the CPP20 intervention, the liver and spleen indexes of mice in all groups were not significantly different from those of the HFHC group. There were no significant differences in the kidney index among all groups.

The liver color of HFHC is light yellow (Figure 4), and the histological analysis results of H&E staining show that the liver has tissue degeneration, which is manifested by the accumulation of fat droplets, cytoplasmic vacuolation, hepatic cord disturbance, and inflammatory cell infiltration. Compared with HFHC, the liver color of CPP20, especially the CPP20M group, was close to NC, and vacuolar degeneration was reduced in CPP20M and CPP20H. The results showed that CPP20 can ameliorate liver damage and tissue lesions caused by a high-fat and high-cholesterol diet.

#### 3.5.2. Effects of CPP20 on Liver Function and Antioxidant Enzymes

Aminotransferase (ALT and AST) is an important indicator of liver injury and necrosis, and the increase of serum aminotransferase activity indicates pathological changes of the liver. The effects of CPP20 on transaminase and antioxidant enzymes in hypercholesterolemic mice are shown in Figure 5. The activities of ALT and AST in the serum and liver of mice fed with a high-fat and high-cholesterol diet were significantly increased (*p* < 0.05). Compared with HFHC, different doses of CPP20 can significantly reduce the levels of ALT and AST in serum and liver (*p* < 0.05). Oxidative stress plays a key role in the occurrence and development of hypercholesterolemia. Compared with NC, the activities of GSH-Px, SOD, and CAT in the serum and liver of HFHC were significantly decreased, and the content of MDA in the liver was significantly increased (*p* < 0.05). Compared with HFHC group, GSH-Px, SOD, and CAT levels of CPP20 and SIM groups, they were increased to varying degrees. The MDA content in the liver of CPP20M and CPP20H was significantly lower than that of HFHC, and the activities of SOD, CAT, and GSH-Px were significantly increased (*p* < 0.05). These results suggest that the intervention of CPP20 can improve the activity of antioxidant enzymes, reduce the content of MDA, and improve the antioxidant capacity of the body.

#### 3.5.3. Antihypercholesterolemic Activity of CPP20 In Vivo

The levels of TC, TG, LDL-C, and TBA in HFHC were significantly increased, while the levels of HDL-C were significantly decreased (*p* < 0.05, Figure 6). Compared with HFHC, the levels of TC, TG, LDL-C, and TBA in the liver and serum of CPP20 groups and SIM were significantly decreased, and the levels of HDL-C were significantly increased (*p* < 0.05).

#### 3.5.4. The mRNA Expression of Antioxidant-Related and Cholesterol Metabolism-Related Gene in Liver

The expression of antioxidant genes and cholesterol homeostasis genes in liver tissues of each group is shown in Figure 7. Compared with NC, the expression levels of SOD, GSH-Px, CAT, and CYP7A1 genes in the liver tissue of HFHC were significantly decreased, while the expression levels of FXR, HMGCR, SREBP-2, and CYP27A1 genes were significantly increased (*p* < 0.05). Different doses of CPP20 can reverse the abnormal gene expression in hypercholesterolemic mice to different degrees, and CPP20M has the best effect.

#### 3.5.5. The Protein Expression of Cholesterol Homeostasis Related of Liver Tissue

Proteins associated with cholesterol metabolism in the liver were determined by western blot (Figure 7). The changing trend of CYP7A1, SREBP-2, and HMGCR was roughly the same as that of mRNA expression. Compared with NC, the expression of CYP7A1 in HFHC was significantly decreased (*p* < 0.05), while the expression of SREBP-2 and HMGCR protein was significantly increased. This trend was reversed by the intake of CPP20, and the effect of CPP20M was significant (*p* < 0.05).

#### 3.5.6. Effects of CPP20 on the SCFAs Production

As shown in Table 6, a high-fat and high-cholesterol diet reduced levels of SCFAs. Positive drug and CPP20 treatment increased the content of SCFAs, and the levels of acetic acid, propionic acid, isobutyric acid, isovaleric acid, and valeric acid were significantly increased in CPP20M (*p* < 0.05).

#### 3.5.7. Effects of CPP20 on the Gut Microbiota

The effects of CPP20 on gut microbes and their functions are shown in Figure 8, Figure 9 and Figure 10. The α diversity is often used to describe the abundance and evenness of microorganisms, which is important for protecting the health of the organism. As shown in Figure 8A, a high-fat and high-cholesterol diet reduced Chao1, Shannon, Simpson, and Pielou_e indexes of mice. After treatment with simvastatin and different doses of CPP20, the above indexes were further changed, but there was no significant difference compared with HFHC. The β diversity is often used to show the degree of variation in species abundance distribution between experimental groups. As shown in Figure 8B,C, the high-fat and high-cholesterol diet resulted in a separation of gut microbes from NC. After the ingestion of CPP20 and simvastatin, intestinal microbes were separated from HFHC to varying degrees.

As shown in Figure 8 and Table 7, the high-fat and high-cholesterol diet changes the intestinal microbial composition of mice, increasing the relative abundance of *Proteobacteria* and *Verrucomicrobacteria* at the phylum level, and decreasing the relative abundance of *Bacteroidetes* and *Firmicutes*. At the family level, the levels of *Desulfovibrionaceae*, *Lachnospiraceae*, *Verrucomicrobiaceae,* and *Erysipelothriaceae* of HFHC were increased, while the levels of *S24-7* were decreased. Compared to NC, the abundance of *Akkermansia_muciniphila* and *Desulfovibrio_C21_c20* in HFHC increased at species levels. After an intake of CPP20, the intestinal microbial composition of mice with a high-fat and high-cholesterol diet changed, and the effect of different doses of CPP20 was different. Compared with HFHC, the abundance of Proteobacteria and Firmicutes decreased in CPP20M, while the abundance of Bacteroidetes and *Verrucomicrobacteria* increased. At the family level, the abundance of *Lachnospiraceae*, *Erysipelothriaceae,* and [*Paraprevotellaceae*] decreased, and the abundance of *S24-7*, *Verrucomicrobiaceae*, *Ruminaceae,* and *Prevotellaceae* increased. At the species level, *Akkermansia_muciniphila* and *Bacteroides_acidifaciens* increased, while *Desulfovibrio_C21_c20* decreased.

LEfSe’s (Figure 9) results showed the dominant bacteria in each group of mice. The dominant bacteria in NC were related to *Bacteroides*, such as *f_S24-7*, *g_*[*Prevotella*], and *f_*[*Paraprevotellaceae*], while the dominant bacteria in HFHC were mostly related to *Proteobacteria*, such as *f_Desulfovibrionales*, *f_Alcaligenaceae,* and *g_Sutterella*, as well as *Atopostipes_suicloacalis* and *Alcaligenes_faecalis*. The predominant bacteria in CPP20M were *s_Akkermansia_mucinphila* of *Verrucomicrobacteria*. The dominant bacteria of SIM were *Firmicutes*-related bacteria, such as *g_Ralstonia*, *g_Turicibacter,* and *f_Turicibacteraceae*.

The results of correlation analysis show that the composition of NC is similar to that of CPP20M and CPP20H but different from that of HFHC, SIM, and CPP20L. NC is positively correlated with *Alistipes_indistinctus*, *Ruminococcus_flavefaciens,* and *Lactobacillus_vaginalis*, CPP20M is positively correlated with *Akkermansia_muciniphila*, *Bacteroides_acidifaciens,* and *Staphylococcus_sciuri*, and HFHC is positively correlated with *Desulfovibrio_C21_c20*, *Bifidobacterium_animalis*, and *Clostridium_aldenense*. Among them, *Akkermansia_muciniphila* was significantly positively correlated with the level of HDL-C and negatively correlated with ALT and CYP27A1, *Clostridium_aldenense* was positively correlated with LDL-C, TC, TG, and FXR and negatively correlated with GSH-Px. *Desulfovibrio_oxamicus* was significantly positively correlated with TC, TG, LDL-C, HMGCR, SREBP-2, and FXR and significantly negatively correlated with GSH-Px and CYP7A1.

The ingestion of CPP20 alters the function of gut microbiota. As shown in Figure 10, compared with the NC, HFHC up-regulated five metabolic pathways, and the CPP20M up-regulated 29 metabolic pathways and down-regulated 21 metabolic pathways. Compared with HFHC, CPP20M down-regulates five metabolic pathways. It is suggested that CPP20 alleviates hypercholesterolemia by altering the intestinal microbiome and its function.

## 4. Discussion

In this study, the cholesterol-lowering activity of CPP20 was investigated in vitro and in vivo. CPP20 used in this experiment was a homogeneous polysaccharide with Mw of 56.282 kDa, obtained by fractional alcohol precipitation. The total carbohydrate content was 67.18 ± 1.22%, the uronic acid content was 25.60 ± 0.37%, and the protein content was 4.21 ± 0.57%. The monosaccharide composed of CPP20 was Glc, Ara, GalA, Gal, Rha, NAG, GlcA, Xyl, GlcN, Fuc, and GlaN, with the molar percentage being 0.296:0.214:0.201:0.170:0.036:0.025:0.020:0.018:0.010:0.008:0.003 mol%.

The BAs are derivatives of cholesterol, and primary BAs exist mainly in the form of BS. It has physiological functions of cholesterol dissolution and fat transfer in the gastrointestinal tract and promotes cholesterol consumption [25]. Therefore, reducing BS levels not only helps to reduce the accumulation of BAs but also accelerates cholesterol metabolism and reduces cholesterol content. Polysaccharides are reported to reduce the dissolution of cholesterol, inhibit its absorption, and reduce the BAs by binding to cholesterol and BS. At the same time, a decrease in BAs leads to a decrease in micellar solution, which in turn impinges on lipid and cholesterol absorption. Furthermore, BAs that bind to polysaccharides are also excreted from the feces, further lowering cholesterol [26]. The cholesterol-lowering and other biological activities of polysaccharides are affected by molecular weight, concentration or dose, linkage type, and molecular properties. The functional groups of polysaccharides include -COH, C=O, -O-, -COOH, and -OH, which attract and remove cholesterol and BAs from the gastrointestinal tract [27]. The results of in vitro experiments showed that CPP20 had the capacity of the cholesterol inhibition rate and the BA binding rate. This may be because the porous structure of CPP20 provides an active site for BA binding, and it has abundant functional groups such as -OH to promote its binding with BAs. On the other hand, the strong BA binding ability of CPP20 may be due to its high uronic acid content and good antioxidant activity. It has been reported that polysaccharide obtained from acid-assisted extraction has the strongest BA binding ability, which may be related to high aldehyde content [28]. Oxidation will lead to a decline in the ability of β-glucan to restrict the flow of BAs [29]. The effect of antioxidant capacity on the adsorption of BAs by polysaccharides may be due to the fact that the molecular groups and structures that affect antioxidant activity also affect the binding ability of BAs [30]. In vitro results of this study are consistent with those previously reported, CPP20 has the best antioxidant capacity, and its ability to absorb BAs is also the strongest. Therefore, CPP20 may exert its cholesterol-lowering activity by forming a network that effectively traps cholesterol and BAs in the aqueous phase, inhibiting cholesterol absorption and increasing BAs excretion [26,31].

The liver index reflects liver function and health, and ALT and AST are the most common serum biomarkers for a variety of liver tissue lesions. A high-fat and high-cholesterol diet can increase the production of ROS and MDA by increasing the levels of enzymes such as ALT and AST, ultimately leading to an increase in circulating cholesterol concentration [32]. In this study, the liver index of CPPM and SIM was higher than that of HFHC, which may be due to the lower body weight of CPPM and SIM, and transaminase levels and H&E results showed significant improvement in CPP20M. Studies have shown that oxidative stress can lead to liver diseases [33,34]. ROS is the main cause of oxidative stress and lipid peroxidation, leading to the depletion of antioxidant enzymes, causing cell damage involved in cholesterol metabolic pathways, and ultimately leading to dyslipidemia and related diseases [34]. It has been reported that polysaccharides inhibit oxidative stress by acting as a natural scrubber of ROS and increasing the levels of antioxidant enzymes (SOD, CAT, GSH-Px), thus producing the potential of lowering blood lipids and cholesterol [35,36]. The results of this study are consistent with the results of previous studies, CPP20 increased the levels of SOD, CAT, and GSH-Px, reduced the level of MDA in the liver, and alleviated hypercholesterolemia by enhancing the antioxidant function.

The liver can excrete cholesterol into the plasma circulation in the form of very low-density lipoprotein (VLDL) and/or LDL. At the same time, the liver releases newly synthesized HDL-C, which removes excess cholesterol from the plasma [37,38]. LDL is a lipoprotein particle that carries cholesterol into peripheral tissue cells and can be oxidized to Ox-LDL, which is found in high levels in patients with hypercholesterolemia and cardiovascular disease [39,40]. LDL-C formed by the LDL and cholesterol is the carrier of TC, and the accumulation of LDL and LDL-C is easy to lead to atherosclerotic plaque lesions. HDL can transport cholesterol from peripheral tissues to the liver for catabolism through the “reverse cholesterol transport” pathway, and high levels of HDL have a protective effect [33]. After the administration of CPP20, the levels of TC, TG, LDL-C, and TBA in the serum and liver of hypercholesterolemic mice are significantly reduced, and the levels of HDL-C are significantly increased.

The endogenous cholesterol required by the body is mainly synthesized in the liver, and HMGCR is the main rate-limiting enzyme for cholesterol synthesis, which is a well-known target for statins [41]. Polysaccharides have been reported to competitively inhibit HMGCR and restrict endogenous cholesterol synthesis [42]. The SREBP-2 transcription factor is related to the overexpression of HMGCR and is involved in the biosynthesis of cholesterol in the liver, and the overexpression of the SREBP-2 transcription factor will lead to elevated cholesterol levels [43]. The results of this experiment are consistent with previous reports; CPP20 reduced cholesterol levels by inhibiting the expression of SREBP-2 and HMGCR and inhibiting cholesterol synthesis [44]. The increased synthesis of BAs stimulates the utilization of cholesterol, which is considered to be the main metabolic pathway of cholesterol catabolism. There are two ways to synthesize BAs from cholesterol metabolism in the liver. One is the classical pathway catalyzed by CYP7A1, and the other is the alternative pathway catalyzed by CYP27A1. The classical pathway is the main pathway of BA synthesis in the human body, producing about 75% of BAs, and CYP7A1 activity is negatively regulated by FXR, while FXR is negatively regulated by free BA content [45]. In this study, CPP20 increased the expression level of CYP7A1, decreased the expression of CYP27A1 and FXR, and lowered cholesterol by promoting the synthesis of BAs (Figure 11).

Some gut bacteria could produce bile brine hydrolase (BSH) to hydrolyze bound BAs to free BAs, thereby increasing BA excretion and reducing total cholesterol and BA levels [46]. Glycine and taurine released in the deconjugation reaction become nutrient sources for the gut microbiome, and the optimal pH for this reaction is about 6 [47]. In this study, the gut microbiota of mice with hypercholesterolemia was significantly different compared to NC. The decrease of α diversity in CPP20, especially in CPP20M, may be related to the bacteriostasis of some BAs (mainly DCA and LCA). *Akkermansia muciniphila*, the dominant bacterium in CPP20M, was reported to have a decreased abundance of mice with metabolic disease, and symptoms of the diseased mice improved after supplementation with *Akkermansia muciniphila* [48]. The intragastric administration of *Akkermansia muciniphila* can significantly improve intestinal mucosal barrier dysfunction and metabolic disorders in mice with a high-fat diet; *Akkermansia muciniphila* can enhance the efficacy of metformin in the treatment of Type 2 diabetes, prevent atherosclerosis, and alleviate metabolic endotoxemia in mice [49,50,51]. The dominant bacteria in HFHC, the members of *Proteobacteria*, are often closely related to the occurrence of some diseases. Among them, *Alcaligens_faecalis* is extensively resistant, and its infection can cause cystitis, diabetes, and pneumonia, among other diseases [52]. *Bacteroides_acidifaciens*, which are positively associated with CPP20M, have been reported to have the potential to treat metabolic diseases such as diabetes and obesity [53]. *Eubacterium* is a producer of butyrate and propionate, and it reduces cholesterol levels by converting cholesterol to coprostanol and regulating BA metabolism. Its levels in the gut microbiota are affected by the amount of dietary fiber in the gut, and both high-protein/high-fat diets reduce their levels [54]. *Subdoligranulum* is almost absent in people with obesity and diabetes but is systematically present in healthy people [55]. The *Staphylococcus sciuri* species is generally considered harmless, and almost all strains of the *Staphylococcus* genus carry BSH, which can promote the production of free BAs [56,57]. *Desulfovibrio C21-20*, which is positively related to HFHC, is often considered a harmful bacterium in the gut and has been linked to inflammatory bowel disease [58]. Among other bacteria positively related to HFHC, *Clostridium_cocleatum* is reported to be enriched in obese mice [59], *Corynebacterium_stationis* and *Psychrobacter_sanguinis* are considered as pathogenic bacteria associated with body infection [59,60,61]. In addition, the intake of CPP20 also increased the level of SCFA in the gut. Butyric acid can inhibit the activity of SREBP-2 and the expression of HMGCR, thereby inhibiting cholesterol synthesis and lowering cholesterol levels. Propionic acid can consume plasma cholesterol by reducing HMGCR activity and inhibiting acetyl-Co reductase, which catalyzes acetic acid to synthesize acetyl-CoA [13,62]. In addition, acetic acid has been reported to reduce serum total cholesterol and triglyceride levels in rats fed a high-cholesterol diet [63]. The production of SCFAs also reduces the pH of the large intestine, promotes the unbinding of conjugated BAs, reduces the solubility of BAs, and reduces the conversion of primary BAs into secondary BAs by the bacterial enzyme 7α-dehydroxylase (associated with cholesterol emulsification), which is associated with cholesterol emulsification [64,65,66]. Therefore, changing the composition of intestinal flora and increasing the level of SCFAs are important ways for CPP20 to relieve hypercholesterolemia.

Compared with NC, the function of increased abundance in HFHC is more related to glycolysis. The PWY0-1479 pathway is associated with an increased incidence of Type 2 diabetes [67]. The function of increased abundance in CPP20M is associated with glycolysis, lipid oxidation, and vitamin production. Among them, PWY-6891 is a synthetic pathway for thiazoles, which are commonly used to treat Type 2 diabetes [67]. Pyridoxal 5′-phosphate is a chemically active form of Vitamin B6, it is a cofactor in more than 160 enzyme activities, and it is closely related to metabolism [68]. Sulfate assimilation and fatty-acid β-oxidation (SO4ASSIM-PWY and FAO-PWY) are associated with SCFA production [69]. Fucose, galactose, uronic acid degradation, and other related pathways (FUCCAT-PWY, GALACTARDEG, and GLUCARGALACTSUPER-PWY) are upregulated, which may be related to the monosaccharide composition of CPP20. CPP20 also facilitated the conversion of pyruvate to butyrate and acidifying fermentation (CENTFERM-PWY and PWY6590). LPS produced by gut microbes has been reported to increase LDL-C and decrease HDL-C, possibly by promoting HMGCA reductase [70]. CPP20M reduced some of the functional pathways associated with LPS production (PWY6478, PWY7315) and reduced propionic acid degradation (PWY5747, PWY0-42), which may relate to reduced levels of gram-negative bacteria in CPP20M. PWY-6629, and PWY-6165 decreased by CPP20, the metabolites may promote the formation of abdominal aortic aneurysms and are positively correlated with E. coli/Shigella [71]. PWY-7347 and SUCSYN-PWY metabolic pathways are highly active in chronic endometritis [72].

## 5. Conclusions

This study first reported the cholesterol-lowering effect of *Cyclocarya paliurus* polysaccharides (CPP20). In vitro, CPP20 had excellent antioxidant activity and can effectively bind BAs and inhibit the dissolution of cholesterol. In vivo, CPP20 enhanced the activity of antioxidant enzymes (CAT, SOD, and GSH-Px) and regulated TC, TG, LDL-C, HDL-C, and TBA levels in the serum and liver to relieve symptoms of hypercholesterolemia. In addition, CPP20 regulated the expression of antioxidant-related and cholesterol-homeostasis genes in the liver, reducing cholesterol levels by promoting the conversion of cholesterol to BAs. CPP20 also regulated cholesterol metabolism by mediating the structure and function of gut microbes and promoting the production of SCFAs. Therefore, these findings suggest that CPP20 is an alternative source for alleviating hypercholesterolemia and protecting cardiovascular health.

## Figures and Tables

**Figure 1 foods-13-02343-f001:**
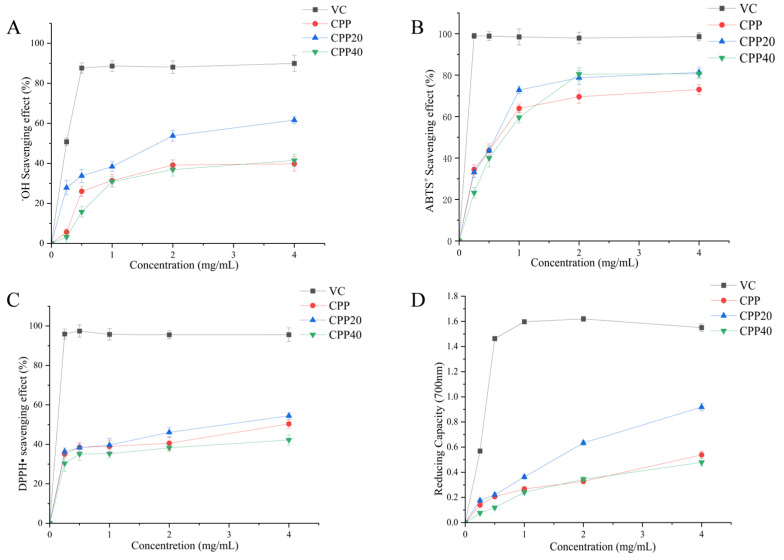
Scavenging effects of *Cyclocarya paliurus* polysaccharides on DPPH•, ABTS+, •OH, and reducing power. The values were presented as M ± SD (*n* = 3). CPP: Crude polysaccharide; CPP20: Polysaccharides obtained by 20% alcohol; CPP40: Polysaccharides obtained by 40% alcohol; VC: Vitamin C. (**A**) •OH Scavenging effect; (**B**) ABTS+ Scavenging effect; (**C**) DPPH• Scavenging effect; (**D**) Reducing Capacity.

**Figure 2 foods-13-02343-f002:**
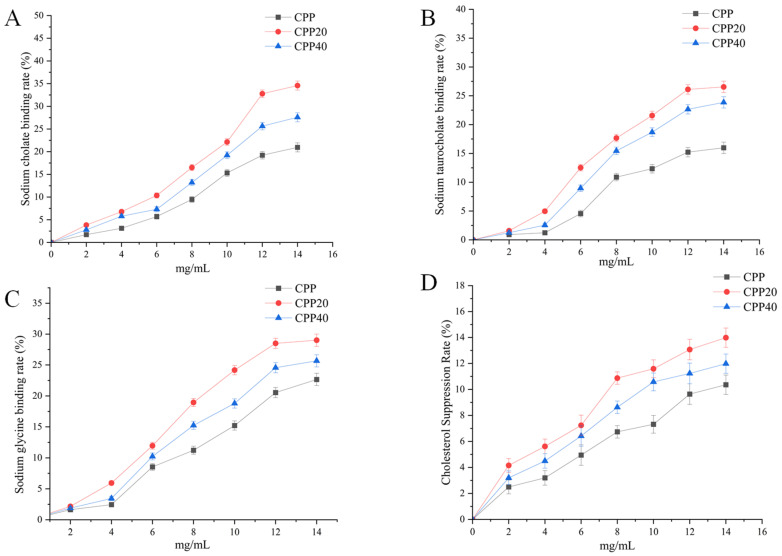
BAs’ binding rate and cholesterol micelle solubility inhibition capacity of *Cyclocarya paliurus* polysaccharides. The values were presented as M ± SD (*n* = 3). CPP: Crude polysaccharide; CPP20: Polysaccharides obtained by 20% alcohol; CPP40: Polysaccharides obtained by 40% alcohol. (**A**) Sodium cholate binding rate; (**B**) Sodium taurocholate binding rate; (**C**) Sodium glycine binding rate; (**D**) Cholesterol Suppression.

**Figure 3 foods-13-02343-f003:**
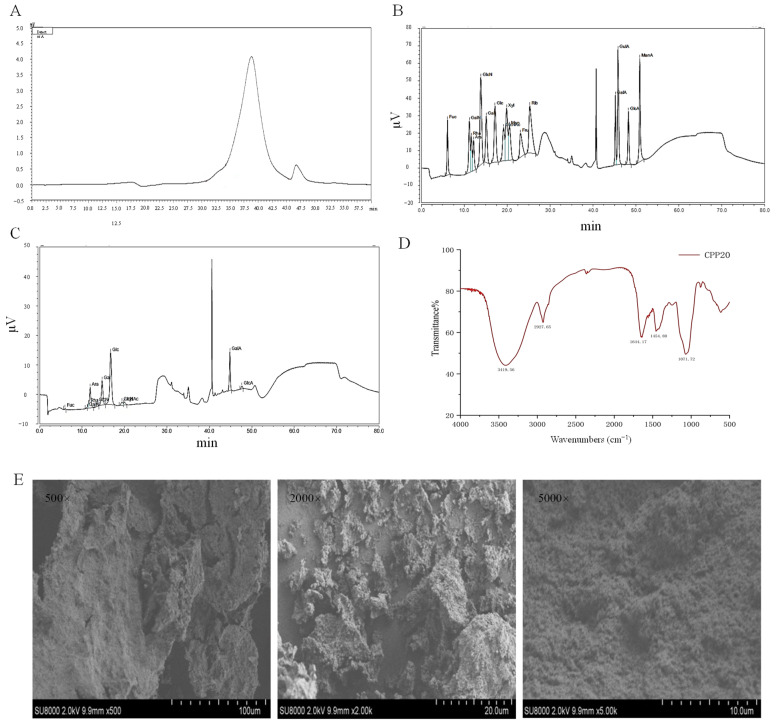
Structure of CPP20. (**A**) Molecular weight; (**B**) Monosaccharide standard; (**C**) Monosaccharide composition of CPP20; (**D**) FT-IR; (**E**) SEM. Glc: glucose; Ara: arabinose; GalA: galacturonic acid; Gal: galactose; Rha: rhamnose; NAG: *N*-Acetyl-D glucosamine; GlcA: glucuronic acid; Xyl: xylose; GlcN: glucosa-mine hydrochloride; Fuc: fucose; GalN: galactosamine hydrochloride.

**Figure 4 foods-13-02343-f004:**
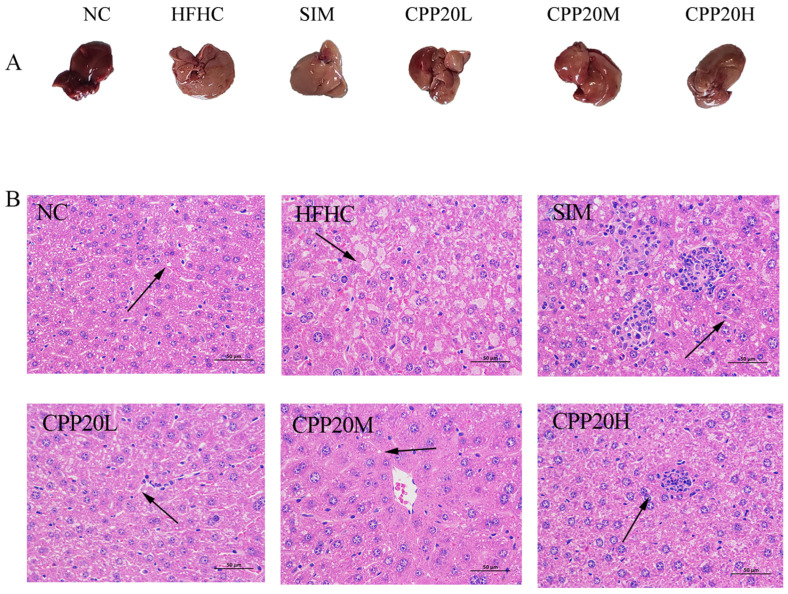
Histopathology of the liver. The arrow points to the lesion. NC: normal group; HFHC: model group; CPP20L: low-dose polysaccharide group; CPP20M: medium-dose polysaccharide group; CPP20H: high-dose polysaccharide group; SIM: positive drug group. (**A**) Liver appearance; (**B**) H&E.

**Figure 5 foods-13-02343-f005:**
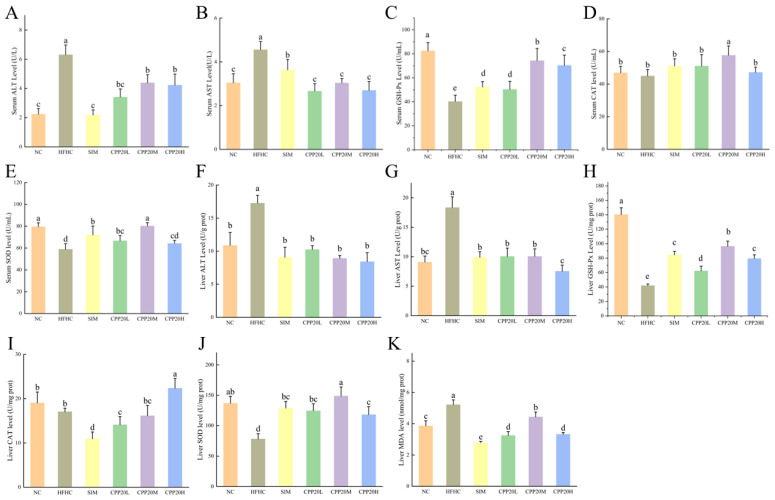
Levels of transaminase and antioxidant enzymes in serum and liver. The values were presented as M ± SD (*n* = 6). On each measure, the difference in letters between the groups indicated significant differences by the letters a–e (*p* < 0.05). NC: normal group; HFHC: model group; CPP20L: low dose polysaccharide group; CPP20M: medium dose polysaccharide group; CPP20H: high dose polysaccharide group; SIM: positive drug group; ALT: alanine transaminase; AST: aspartate aminotransferase; GSH-Px: glutathione peroxidase; CAT: catalase; SOD: superoxide dismutase; MDA: Malondialdehyde. (**A**) Serum ALT level; (**B**) Serum AST level; (**C**) Serum GSH-Px level; (**D**) Serum CAT level; (**E**) Serum SOD level; (**F**) Liver ALT level; (**G**) Liver AST level; (**H**) Liver GSH-Px level; (**I**) Liver CAT level; (**J**) Liver SOD level; (**K**) Liver MDA level.

**Figure 6 foods-13-02343-f006:**
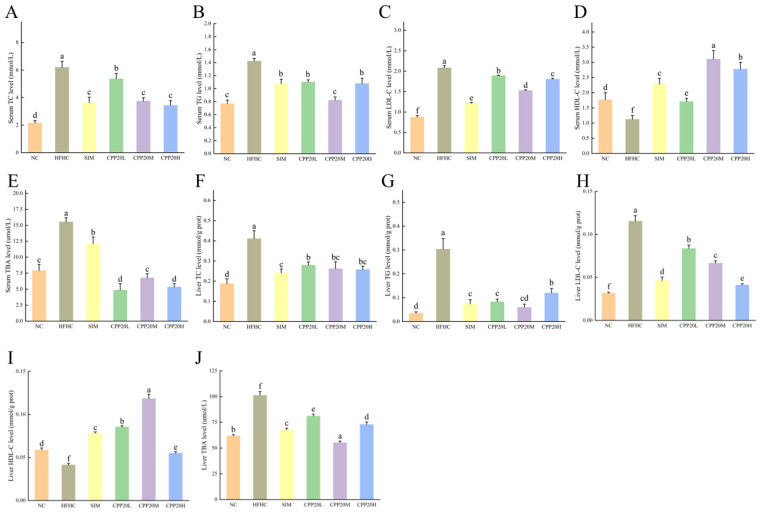
Levels of TC, TG, LDL-C, HDL-C, and TBA in serum and liver. The values were presented as M ± SD (*n* = 6). On each measure, the difference in letters between the groups indicated significant differences by the letters a–f (*p* < 0.05). NC: normal group; HFHC: model group; CPP20L: low-dose polysaccharide group; CPP20M: medium-dose polysaccharide group; CPP20H: high-dose polysaccharide group; SIM: positive drug group; TC: total cholesterol; TG: triglyceride; LDL-C: low-density lipoprotein cholesterol; HDL-C: high-density lipoprotein cholesterol; TBA: total bile acid. (**A**) Serum TC level; (**B**) Serum TG level; (**C**) Serum LDL-C level; (**D**) Serum HDL-C level; (**E**) Serum TBA level; (**F**) Liver TC level; (**G**) Liver TG level; (**H**) Liver LDL-C level; (**I**) Liver HDL-C level; (**J**) Liver TBA level.

**Figure 7 foods-13-02343-f007:**
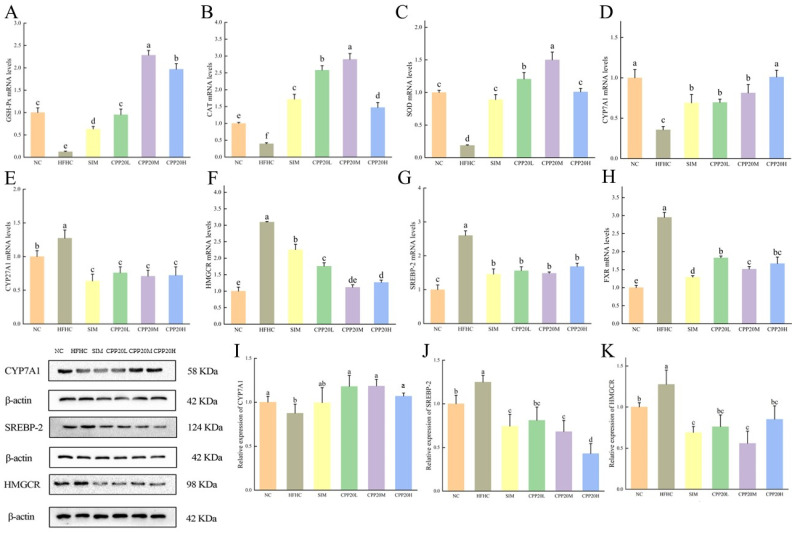
Expression of genes and proteins related to antioxidant and BAs metabolism. The values were presented as M ± SD (*n* = 3). On each measure, the difference in letters between the groups indicated significant differences by the letters a–f (*p* < 0.05). NC: normal group; HFHC: model group; CPP20L: low-dose polysaccharide group; CPP20M: medium-dose polysaccharide group; CPP20H: high-dose polysaccharide group; SIM: positive drug group; GSH-Px: glutathione peroxidase; CAT: catalase; SOD: superoxide dismutase; CYP7A1: cholesterol 7-alpha hydroxylase; CYP27A1: mitochondrial sterol 27-hydroxylase; HMGCR: 3-hydroxy-3-methylglutaryl-CoA reductase; SREBP-2: sterol-regulatory element binding protein 2; FXR: farnesoid X receptor. (**A**) GSH-Px mRNA levels; (**B**) CAT mRNA levels; (**C**) SOD mRNA levels; (**D**) CYP7A1 mRNA levels; (**E**) CYP27A1 mRNA levels; (**F**) HMGCR mRNA levels; (**G**) SREBP-2 mRNA levels; (**H**) FXR mRNA levels; (**I**) Relative expression of CYP7A1; (**J**) Relative expression of SREBP-2; (**K**) Relative expression of HMGCR.

**Figure 8 foods-13-02343-f008:**
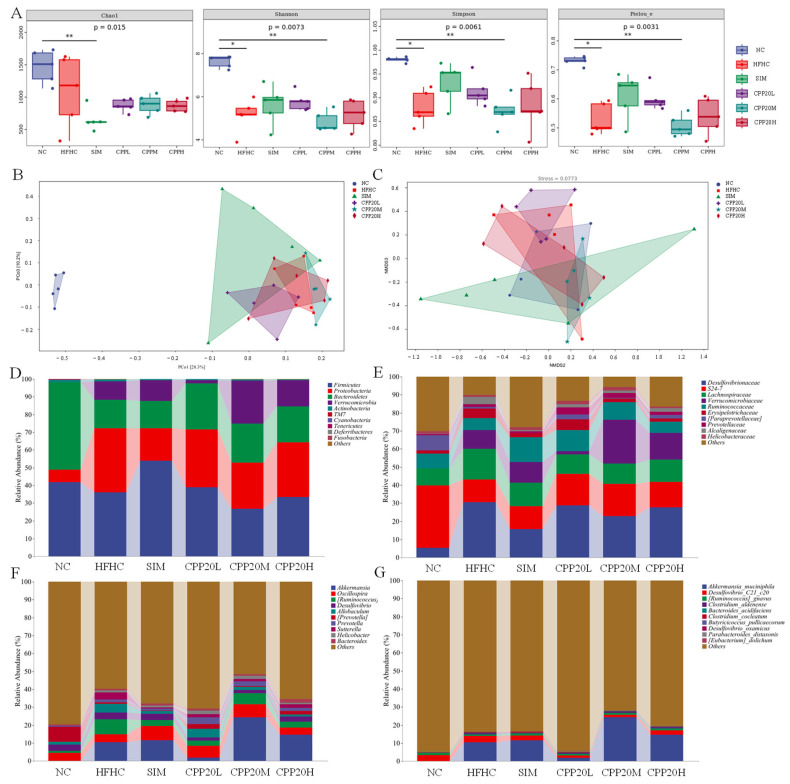
α, β-diversity analysis and microbial community composition of each group. (**A**) α-diversity; (**B**) PCoA; (**C**) NMDS; (**D**–**G**) Relative abundance of microflora at phylum, family, genus, and species level, respectively. NC: normal group; HFHC: model group; CPP20L: low-dose polysaccharide group; CPP20M: medium-dose polysaccharide group; CPP20H: high-dose polysaccharide group; SIM: positive drug group. * *p* < 0.05, ** *p* < 0.01.

**Figure 9 foods-13-02343-f009:**
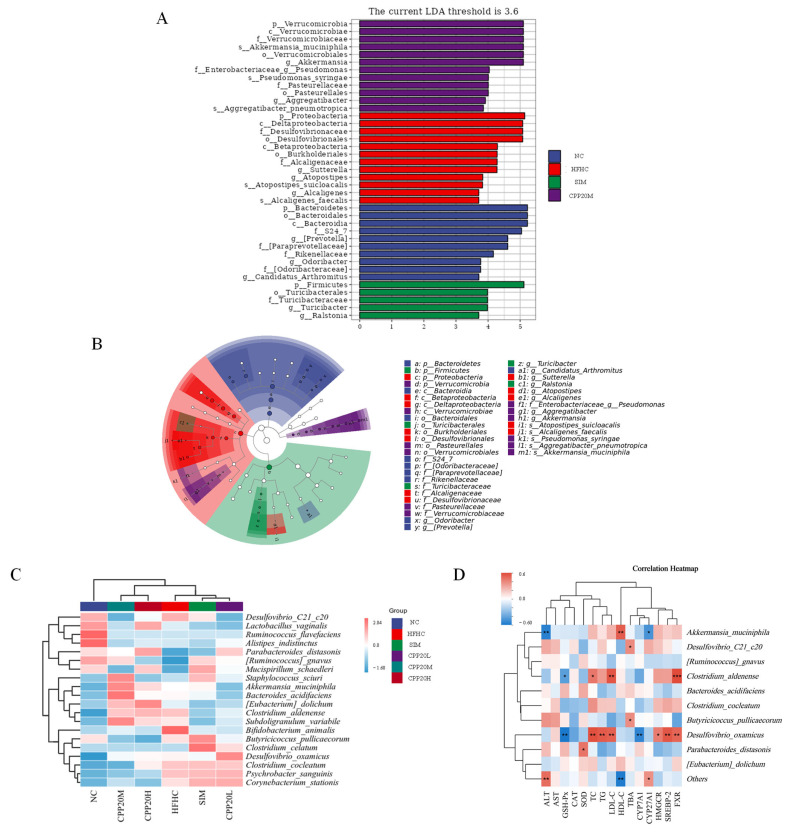
Differentiation and correlation analysis of microorganisms in each group. (**A**) LEfSe analysis; (**B**) Histogram and chart; (**C**) Heatmap analysis at the species level; (**D**) Correlation analysis (*n* = 5). NC: normal group; HFHC: model group; CPP20L: low-dose polysaccharide group; CPP20M: medium-dose polysaccharide group; CPP20H: high-dose polysaccharide group; SIM: positive drug group. * *p* < 0.05, ** *p* < 0.01, *** *p* < 0.01.

**Figure 10 foods-13-02343-f010:**
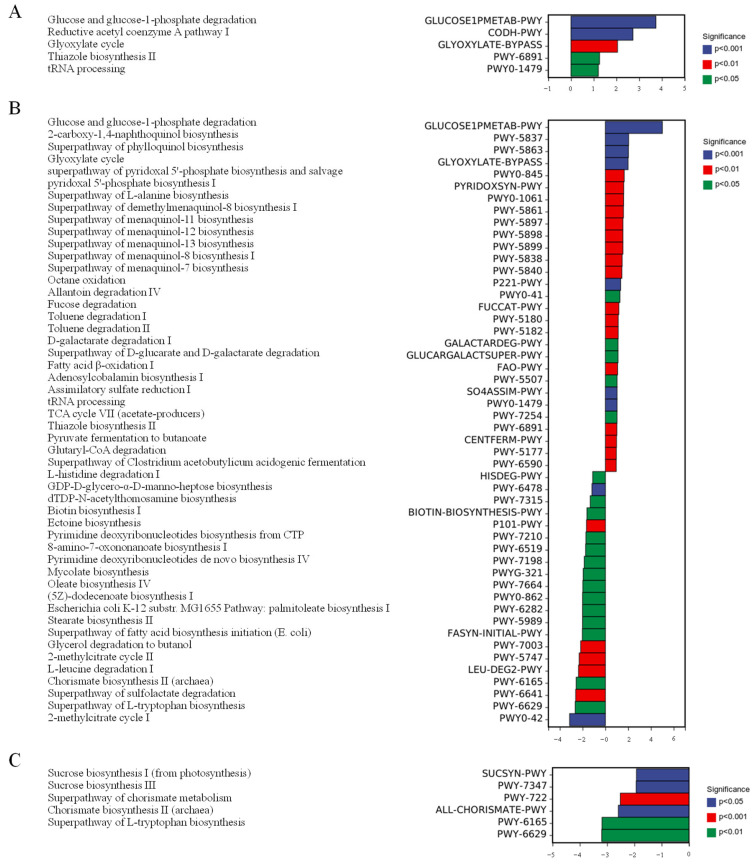
Differential analysis of intestinal microbial function. (**A**) HFHC vs. NC; (**B**) CPP20M vs. NC; (**C**) CPP20M vs. HFHC. NC: normal group; HFHC: model group; CPP20L: low-dose polysaccharide group; CPP20M: medium-dose polysaccharide group; CPP20H: high-dose polysaccharide group; SIM: positive drug group.

**Figure 11 foods-13-02343-f011:**
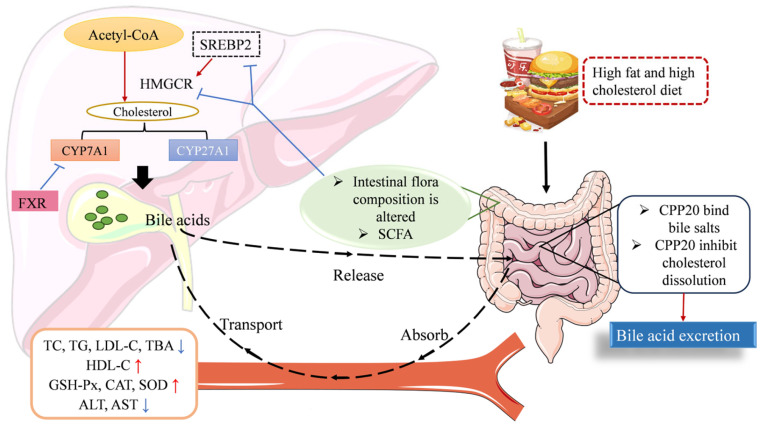
Mechanism diagram of CPP20 regulating cholesterol metabolism. TC: total cholesterol; TG: triglyceride; LDL-C: low-density lipoprotein cholesterol; HDL-C: high-density lipoprotein cholesterol; TBA: total bile acid; GSH-Px: glutathione peroxidase; CAT: catalase; SOD: superoxide dismutase; ALT: alanine transaminase; AST: aspartate aminotransferase; CYP7A1: cholesterol 7-alpha hydroxylase; CYP27A1: mitochondrial sterol 27-hydroxylase; HMGCR: 3-hydroxy-3-methylglutaryl-CoA reductase; SREBP2: sterol-regulatory element binding protein 2; FXR: farnesoid X receptor; SCFA: Short-chain fatty acid.

**Table 1 foods-13-02343-t001:** Composition of high-fat and high-cholesterol diet feed and normal diet.

Feed Composition	Normal Diet (g/kg)	High Fat and High Cholesterol Diet (g/kg)
Casein	140	140
Corn starch	465.692	459.442
Saccharose	100	100
Maltodextrin	155	155
Soybean oil	40	40
Cellulose	50	50
Choline bitartrate	2.5	2.5
Mineral mixture	35	35
Vitamin mixture	10	10
L-cystine	1.8	1.8
Tert-butylhydroquinone	0.008	0.008
Cholesterol	-	5
Cholate	-	1.25

**Table 2 foods-13-02343-t002:** Primer sequence.

Primer	Forward (5′→3′)	Reverse (5′→3′)
β-actin	5′-GCTCTGGCTCCTAGCACCAT-3′	5′-GCCACCGATCCACACAGAGT-3′
SREBP-2	5′-CAGCTGGATCCTCCCAAAGA-3′	5′-CTCAGAACGCCAGACTTGTC-3′
HMGCR	5′-GAGCAGCGACATCATCATCC-3′	5′-GGCCAGCAATACCCAGAATG-3′
CYP7A1	5′-ATCAGGAGCCCTGAAGCAAT-3′	5′-TCTTGGCCAGCACTCTGTAA -3′
CYP27A1	5′-ACCGATGGCTGAGGAAGAAA-3′	5′-TACCAGCCTTGACAGCATCA-3′
FXR	5′-GGGATGAGCTGTGTGTTGTC-3′	5′-ACACGGCGTTCTTGGTAATG-3′
SOD	5′-CGGTGAACCAGTTGTGTTGT-3′	5′-CCCATACTGATGGACGTGGA -3′
CAT	5′-ACATGGTCTGGGACTTCTGG-3′	5′-ACTGCCTCTCCATCTGCATT-3′
GSH-Px	5′-GGGACCCTGAGACTTAGAGC-3′	5′-AATCCGTCAGCAATCATCC-3′

**Table 3 foods-13-02343-t003:** Chemical composition of *Cyclocarya paliurus* polysaccharides.

Sample	Total Carbohydrate (%)	Uronic Acid (%)	Protein (%)
CPP	63.93 ± 1.07 b	20.90 ± 0.43 b	7.44 ± 0.45 a
CPP20	67.18 ± 1.22 a	25.60 ± 0.37 a	4.21 ± 0.57 b
CPP40	66.33 ± 0.93 ab	24.58 ± 0.21 a	3.26 ± 0.28 c

Note: The values were presented as M ± SD (*n* = 3). On each measure, the difference in letters between the groups indicated significant differences by the letters a–c (*p* < 0.05). CPP: Crude polysaccharide; CPP20: Polysaccharides obtained by 20% alcohol; CPP40: Polysaccharides obtained by 40% alcohol.

**Table 4 foods-13-02343-t004:** The weight of the mice during the experiment.

	Week	0	5	6	7	8	9	10
Group								
NC		19.19 ± 0.15 a	21.84 ± 0.65 b	21.90 ± 0.90 b	21.99 ± 1.30 c	22.04 ± 0.62 d	22.26 ± 0.43 d	22.34 ± 0.52 d
HFHC		19.11 ± 0.26 a	25.77 ± 0.56 a	26.72 ± 0.69 a	26.78 ± 0.80 a	26.87 ± 1.14 a	26.99 ± 0.60 a	27.15 ± 0.53 a
SIM		19.18 ± 0.24 a	25.89 ± 0.51 a	26.46 ± 0.54 a	24.65 ± 0.54 b	24.35 ± 0.33 c	23.18 ± 0.54 cd	22.51 ± 0.78 d
CPP20L		19.13 ± 0.50 a	25.79 ± 0.83 a	26.38 ± 0.91 a	26.00 ± 0.73 ab	25.69 ± 0.57 b	25.43 ± 0.95 b	24.73 ± 0.89 b
CPP20M		19.14 ± 0.58 a	25.64 ± 0.86 a	26.30 ± 1.44 a	25.05 ± 1.46 b	24.61 ± 0.93 bc	24.36 ± 1.36 bc	23.12 ± 0.95 cd
CPP20H		19.16 ± 0.34 a	25.65 ± 0.83 a	26.75 ± 0.16 a	25.30 ± 0.68 b	24.91 ± 0.85 bc	24.82 ± 1.02 b	23.88 ± 0.98 bc

Note: The values were presented as M ±SD (*n* = 6). On each measure, the difference in letters between the groups indicated significant differences by the letters abcd (*p* < 0.05). NC: normal group; HFHC: model group; CPP20L: low-dose polysaccharide group; CPP20M: medium-dose polysaccharide group; CPP20H: high-dose polysaccharide group; SIM: positive drug group.

**Table 5 foods-13-02343-t005:** Organ index of each group of mice.

Group	Liver Index	Spleen Index	Kidney Index
NC	3.57 ± 0.57 c	0.23 ± 0.02 d	1.25 ± 0.13 a
HFHC	5.90 ± 0.29 b	0.41 ± 0.01 b	1.16 ± 0.07 a
SIM	6.66 ± 0.35 a	0.52 ± 0.04 a	1.23 ± 0.16 a
CPP20L	5.78 ± 0.66 b	0.42 ± 0.04 b	1.24 ± 0.11 a
CPP20M	6.40 ± 0.67 ab	0.42 ± 0.04 b	1.30 ± 0.17 a
CPP20H	5.98 ± 0.42 b	0.37 ± 0.05 c	1.15 ± 0.07 a

Note: The values were presented as M ± SD (*n* = 6). On each measure, the difference in letters between the groups indicated significant differences by the letters a–d (*p* < 0.05). NC: normal group; HFHC: model group; CPP20L: low-dose polysaccharide group; CPP20M: medium-dose polysaccharide group; CPP20H: high-dose polysaccharide group; SIM: positive drug group.

**Table 6 foods-13-02343-t006:** SCFA levels of mice in each group.

	Acetic Acid	Propionic Acid	Isobutyric Acid	Butyric Acid	Isovaleric Acid	Valeric Acid
NC	36.23 ± 3.71 a	7.55 ± 0.78 ab	1.43 ± 0.18 a	4.88 ± 0.45 a	1.45 ± 0.20 a	1.62 ± 0.46 a
HFHC	18.90 ± 1.97 c	5.01 ± 0.76 d	0.55 ±0.07 c	2.53 ± 0.43 b	0.55 ± 0.09 d	0.56 ± 0.07 c
SIM	25.05 ± 3.16 b	5.69 ± 0.89 cd	0.84 ± 0.19 b	4.87 ± 1.68 a	0.82 ± 0.22 bc	0.91 ± 0.23 bc
CPP20L	22.65 ± 1.75 b	4.95 ± 0.91 d	0.77 ± 0.26 bc	3.05 ± 0.87 b	0.65 ± 0.17 cd	0.94 ± 0.22 bc
CPP20M	25.66 ± 1.99 b	7.82 ± 0.70 a	0.93 ± 0.12 b	3.29 ± 0.44 b	0.92 ± 0.19 b	1.18 ± 0.06 b
CPP20 H	25.35 ± 0.68 b	6.58 ± 0.59 bc	0.70 ± 0.06 bc	2.97 ± 0.64 b	0.72 ± 0.08 bcd	1.04 ± 0.29 b

Note: The values were presented as M ± SD (*n* = 5). On each measure, the difference in letters between the groups indicated significant differences by the letters a–d (*p* < 0.05). NC: normal group; HFHC: model group; CPP20L: low-dose polysaccharide group; CPP20M: medium-dose polysaccharide group; CPP20H: high-dose polysaccharide group; SIM: positive drug group.

**Table 7 foods-13-02343-t007:** Phylum, genus, and species level abundance.

Group	NC	HFHC	SIM	CPP20L	CPP20M	CPP20H
Phylum						
*Firmicutes*	41.87 ± 5.90 b	35.99 ± 4.06 bc	53.97 ± 12.46 a	39.02 ± 5.60 b	26.67 ± 3.72 c	33.47 ± 8.92 bc
*Proteobacteria*	6.92 ± 1.85 d	36.22 ± 6.97 a	18.32 ± 6.34 c	32.63 ± 7.35 ab	26.20 ± 6.25 bc	30.72 ± 5.70 ab
*Bacteroidetes*	49.41 ± 6.16 a	16.00 ± 4.90 b	15.43 ± 6.13 b	25.89 ± 6.97 b	22.00 ± 5.02 b	20.46 ± 11.61 b
*Verrucomicrobia*	0.00 ± 0.00 c	10.47 ± 12.88 abc	11.54 ± 10.58 abc	1.77 ± 1.80 bc	24.34 ± 7.59 a	14.73 ± 12.50 ab
*Actinobacteria*	1.27 ± 0.57 a	1.12 ± 0.51 ab	0.68 ± 0.43 abc	0.59 ± 0.24 bc	0.55 ± 0.19 bc	0.41 ± 0.21 c
*TM7*	0.35 ± 0.21 a	0.00 ± 0.00 b	0.00 ± 0.00 b	0.00 ± 0.00 b	0.00 ± 0.00 b	0.00 ± 0.00 b
Family						
*Desulfovibrionaceae*	5.31 ± 1.78 c	30.54 ± 7.26 a	15.82 ± 6.66 b	28.71 ± 7.32 a	22.84 ± 6.73 ab	27.58 ± 7.86 a
*S24-7*	34.46 ± 4.33 a	12.48 ± 3.55 b	12.48 ± 2.78 b	17.40 ± 5.72 b	17.86 ± 6.02 b	14.20 ± 4.65 b
*Lachnospiraceae*	9.49 ± 1.51 b	17.00 ± 4.73 a	13.04 ± 6.39 ab	10.81 ± 3.66 ab	11.21 ± 2.65 ab	12.36 ± 2.01 ab
*Verrucomicrobiaceae*	0.00 ± 0.00 c	10.47 ± 12.88 abc	11.54 ± 10.58 abc	1.77 ± 1.80 bc	24.34 ± 7.59 a	14.73 ± 12.50 ab
*Ruminococcaceae*	8.19 ± 1.70 ab	6.68 ± 1.82 b	13.59 ± 6.64 a	11.87 ± 3.58 ab	9.70 ± 4.05 ab	6.30 ± 1.86 b
*Erysipelotrichaceae*	1.77 ± 1.76 b	5.14 ± 1.92 ab	2.13 ± 1.46 b	5.78 ± 4.17 a	2.01 ± 1.39 b	1.63 ± 1.21 b
*[Paraprevotellaceae]*	8.30 ± 3.69 a	0.89 ± 0.69 b	0.13 ± 0.07 b	2.70 ± 1.92 b	0.43 ± 0.31 b	1.99 ± 3.03 b
*Prevotellaceae*	0.72 ± 0.59 b	1.49 ± 1.08 b	1.08 ± 1.35 b	3.89 ± 1.50 a	2.56 ± 1.45 ab	1.74 ± 1.73 b
*Alcaligenaceae*	0.22 ± 0.16 b	4.18 ± 1.61 a	0.72 ± 0.77 b	1.76 ± 1.28 b	4.12 ± 1.21 b	2.00 ± 2.39 b
Genus						
*Akkermansia*	0.00 ± 0.00 c	10.47 ± 12.88 abc	11.54 ± 10.58 abc	1.77 ± 1.80 bc	24.34 ± 7.59 a	14.73 ± 12.50 ab
*[Ruminococcus]*	1.19 ± 0.52 b	8.34 ± 6.33 a	3.37 ± 2.58 b	2.68 ± 2.40 b	6.18 ± 2.27 ab	3.33 ± 1.35 b
*Allobaculum*	1.57 ± 1.64 ab	4.79 ± 1.368 a	1.64 ± 1.43 ab	4.78 ± 3.63 a	1.87 ± 1.46 ab	1.26 ± 1.26 b
*[Prevotella]*	8.30 ± 3.69 a	0.89 ± 0.69 b	0.13 ± 0.07 b	2.70 ± 1.92 b	0.43 ± 0.31 b	1.99 ± 3.03 b
*Prevotella*	0.57 ± 0.51 b	1.45 ± 1.06 b	1.08 ± 1.34 b	3.80 ± 1.42 a	2.55 ± 1.45 ab	1.72 ± 1.71 b
*Sutterella*	0.22 ± 0.16 b	4.13 ± 1.61 a	0.68 ± 0.77 b	1.76 ± 1.29 b	1.41 ± 1.21 b	1.96 ± 2.34 b
Species						
*Akkermansia_muciniphila*	0.00 ± 0.00 c	10.47 ± 12.88 abc	11.54 ± 10.58 abc	1.77 ± 1.80 bc	24.34 ± 7.59 a	14.73 ± 12.50 ab
*Clostridium_aldenense*	0.00 ± 0.00 b	0.61 ± 0.42 a	0.19 ± 0.29 ab	0.34 ± 0.35 ab	0.60 ± 0.05 a	0.67 ± 0.52 a
*Butyricicoccus_pullicaecorum*	0.11 ± 0.03 ab	0.11 ± 0.15 ab	0.19 ± 0.14 a	0.08 ± 0.09 ab	0.03 ± 0.02 b	0.01 ± 0.01 b
*Desulfovibrio_oxamicus*	0.01 ± 0.01 b	0.10 ± 0.07 ab	0.08 ± 0.03 ab	0.17 ± 0.08 a	0.09 ± 0.04 ab	0.09 ± 0.07 ab

Note: Data are presented as M ± SD (*n* = 5). On each measure, the difference in letters between the groups indicated significant differences by the letters abcd (*p* < 0.05). NC: normal group; HFHC: model group; CPP20L: low-dose polysaccharide group; CPP20M: medium-dose polysaccharide group; CPP20H: high-dose polysaccharide group; SIM: positive drug group.

## Data Availability

The data presented in this study are available on request from the corresponding author. The data are not publicly available due to privacy restrictions.

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
