# Peer review of "Cholesterol-Lowering Effect of Polysaccharides from *Cyclocarya paliurus* In Vitro and in Hypercholesterolemia Mice"

_foods, 2024, doi:10.3390/foods13152343_

Round 1
Reviewer 1 Report
Comments and Suggestions for Authors
This study attempted to explore the effect of Cyclocarya paliurus polysaccharides on hypercholesterolemia through in vivo and in vitro experiments to explore the mechanism of action. The theme of the paper is important for the audience.
minor comments:
I suggest including the meaning of the acronyms under all Figures to make easier the interpretation of the Figures
Line 98: How many grams of leaves were dried?
Line 103: How many grams were obtained by freeze-drying
Line 104: please, mention the obtained volume of CPP20 and CPP40, after precipitation with 20% and 40% ethanol
If possible, show all the Figures online because as they appear on the paper are difficult to understand.
Comments on the Quality of English Language
Minor editing of English language required
Author Response
Comments and Suggestions for Authors 1
This study attempted to explore the effect of Cyclocarya paliurus polysaccharides on hypercholesterolemia through in vivo and in vitro experiments to explore the mechanism of action. The theme of the paper is important for the audience.
minor comments:
I suggest including the meaning of the acronyms under all Figures to make easier the interpretation of the Figures
Response: Thank you for your advice. In order to better understand, we have added the meaning of relevant abbreviations in the notes of the revised manuscript.
Line 98: How many grams of leaves were dried?
Response: Thank you for your good question. In this study, we used 10 kg of dried Cyclocarya paliurus leaves for polysaccharide extraction.
Line 103: How many grams were obtained by freeze-drying
Response: Thank you for your good question. We obtained 350.48 g crude polysaccharide by freeze-drying.
Line 104: please, mention the obtained volume of CPP20 and CPP40, after precipitation with 20% and 40% ethanol
Response: Thank you for your good question. We have added relevant information to the revised manuscript.
If possible, show all the Figures online because as they appear on the paper are difficult to understand.
Response: Thank you for your advice. After the discussion of the authors, some changes have been made to the figures and tables in the revised manuscript.
Comments on the Quality of English Language
Minor editing of English language required
Response: Thank you for your advice. We have checked the grammar problems in the whole text and corrected them.
Reviewer 2 Report
Comments and Suggestions for Authors
This manuscript studies the cholesterol-lowering effect of polysaccharides from Cyclocarya paliurus in vitro and in vivo, in hypercholesterolemia mice and aims to explore the effect of the studied polysaccharides on hypercholesterolemia through in vivo and in vitro experiments to explore their mechanism of action.
Please consider the following suggestions:
Lines 154-156: Please insert the methods used for the in vitro antioxidant activity assays together with the modifications mentioned.
Please insert the Ethics Committee approval registration number for your study.
Please make sure that the words Cyclocarya paliurus, in vivo and in vitro are written in italic form all over the manuscript.
Comments on the Quality of English Language
Minor editing of English language required
Author Response
Comments and Suggestions for Authors 2
This manuscript studies the cholesterol-lowering effect of polysaccharides from Cyclocarya paliurus in vitro and in vivo, in hypercholesterolemia mice and aims to explore the effect of the studied polysaccharides on hypercholesterolemia through in vivo and in vitro experiments to explore their mechanism of action.
Please consider the following suggestions:
Lines 154-156: Please insert the methods used for the in vitro antioxidant activity assays together with the modifications mentioned.
Response: Thank you for your advice. We have added details of in vitro antioxidant experiments to the revised manuscript.
Please insert the Ethics Committee approval registration number for your study.
Response: Thank you for your advice. We provide relevant information in the returned manuscript.
Please make sure that the words Cyclocarya paliurus, in vivo and in vitro are written in italic form all over the manuscript.
Response: Thank you for your advice. We checked and corrected all font formats in the revised manuscript.
Comments on the Quality of English Language
Minor editing of English language required
Response: Thank you for your advice. We have checked the grammar problems in the whole text and corrected them.